# Reflections on the Role of Differentiation Processes in Forming Behavioral Phenotypes: Can These Processes Replace the Concepts of Plastic Phenotype and Reversible Plastic Phenotype?

**DOI:** 10.3390/biology14020187

**Published:** 2025-02-12

**Authors:** Pilar Chiappa

**Affiliations:** Departamento de Etología, Instituto Nacional de Psiquiatría Ramón de la Fuente Muñiz, Camino a Xochimilco 101, Col. San Lorenzo Huipulco, alc. Tlalpan, Mexico City 14370, Mexico; pilar.chiappa@gmail.com; Tel.: +52-5541612470

**Keywords:** behavioral plasticity, human neoteny, adult neuronal differentiation

## Abstract

This essay presents two lines of argument to suggest that the extension into adulthood of specific processes of phenotypic differentiation that are typical of early development underlies the evolution of cognition. The first of these lines is organized around (1) human neoteny (referring to the retention of juvenile characteristics into adulthood due to developmental slowdown during evolution) and its relationship to cognition, and (2) comparative analyses of the correlative evolution of cognitive and developmental processes. The second line argues that the formation of all phenotypes (i.e., rigid, plastic, and reversible plastic) involves processes of interaction between a genetic entity and its environment that occur through successive morpho-functional differentiations, and focuses on the development of cognitive–behavioral phenotypes from neural differentiation processes that occur during their formation.

## 1. Introduction

### 1.1. The Evolution of Developmental and Cognitive Performance

#### 1.1.1. Human Development Has Slowed Down During Evolution

Research comparing various aspects of development shows that human development is comparatively slower than that of the great apes, indicating that the last common ancestor we share with them had faster development than we do or that our developmental pattern has slowed down in the last few million years of evolution [1]. This evolutionary change from the ancestral developmental pattern is called neoteny, a type of heterochrony (i.e., changes in the timing of the appearance of a trait relative to the life span of an organism and changes in the rate of development of the trait relative to the ancestral trait) [1]. Human neoteny is evident in various characteristics, including morphological (as a flattened and broad face), physiological (as a weaker secretion of stress hormones) [2], behavioral (as a prolonged period of dependence after weaning), and cognitive (as an extended learning period), among others [1].

Since the proposition of this heterochronic concept, neoteny has caught the attention of many researchers in evolutionary studies and discussions. In the early decades of the 20th century, developmental studies on several species of primates [3,4,5] showed that in infants, for instance, the position of the foramen magnum does not vary significantly between species and that this location is like that of adult humans. More recent studies confirm these findings, showing that in chimpanzees and bonobos, the position of this structure changes during development in comparison to humans [6]. Simply put, the adult human skull retains the morphological and positional characteristics of primate infants.

From an evolutionary point of view, during the rest of the 20th century, the slowdown or retardation process of human development imposed the need to analyze what the adaptive benefits and costs associated with this trait would have been [7]. The first explanation for the adaptive function of human neoteny was presented in 1950 [8]. In this work, Lorenz explained the concept of the innate releasing mechanism as an adaptation, allowing organisms to react adaptively to biologically relevant situations without the need for prior experience. He discussed the social function of this adaptation in humans, considering the reaction of tenderness toward infant faces as the driving force for the evolution of such a trait. Accordingly, some specific physical characteristics—such as a large head; preponderance of the skull; large low-set eyes; profoundly bulging cheeks; thick, short limbs; elastic consistency; and clumsy movements—have led the morphology of adult human beings to resemble that of an infant, and imitations of this morphology (specifically, dolls, stuffed animals, or cartoons) are considered cute and attractive enough to inhibit the aggressive impulses of others. Lorenz hypothesized that “[the] quality that [a human] has of being an unfinished being is a gift that we must thank neoteny itself for. However, for its part, neoteny is probably a consequence of human domestication” [8]. In this view, human neoteny would have had the adaptive function of reducing intragroup aggression. Lorenz did not claim that behavioral neoteny had been directly selected in any way. Still, he proposed that the relationship between neoteny and cognition would be a fortunate byproduct of human domestication as an evolutionary selective process diminishing intragroup aggression.

Decades later, Stephen Jay Gould also addressed the essential role of heterochrony in human evolution. However, he shifted the focus of selective forces from the morphological domain to the behavioral: “Neoteny has been a (probably the) major determinant of human evolution. […] Retardation as a life-history strategy for longer learning and socialization may be far more important in human evolution than any of its morphological consequences.” [1]. With this hypothesis, Gould opened the possibility for considering the behavioral aspect of human neoteny as the primary target of selective processes, leaving the morphological aspect as a byproduct. He also proposed that the evolution of development is an effective mode of creating complementarity between the organism and its environment through cognitive behavior. This proposal has two theoretical advantages: First, it releases the mutational aspects of the genetic systems from the responsibility of producing the sequence of mutations necessary to obtain preprogrammed cognitive behaviors at the right time in evolutionary human history [9]. Second, due to the emphasis it places on the analysis of development, it allows for the exploration of explanations in the proximate causes of cognitive behavior, making it possible to explain the evolutionary convergences in cognitive behavior found in several taxonomic groups that are phylogenetically and ecologically distant [10].

#### 1.1.2. Evolutionary Studies of Cognitive Behavior

Cognitive behavior depends on experiential phenomena, such as learning, memory, perception, attention, categorization, and motor control [11]. Cognitive behavior is fundamental in explaining human evolution [12]. Two issues affect studies of the evolution of cognitive behavior.

First, behavior is like any other organismal trait, and it must be studied using the same theoretical and methodological approaches as other traits, such as morphological ones [13]. However, evolutionary science has experienced significant theoretical–methodological transformations [14] that have particularly affected the understanding of cognitive behavior. Such is the case in philosophical discussions of the relationship between genotype and phenotype [15], which is made even more difficult by the so-called plasticity that distinguishes the cognitive–behavioral phenotype [16].

Second, the study of the evolution of cognitive behavior has been impregnated by discussions of how complementarity between cognitive systems and way of life are achieved. There are currently three views on the evolution of cognitive behavior [17]. First, the so-called general process view considers that general cognitive systems evolved because they facilitate behavior organization in different environmental facets (for instance, the social and physical). The other view, called the ecological view, looks for cognitive systems that selection processes favor because they allow behavioral solutions to particular and recurring environmental problems in the evolution of a species. Along these lines, for example, the hypotheses of social intelligence [18] and foraging [19] emerged, in which the impositions of social life or foraging, respectively, would have facilitated the evolution of cognitive abilities. However, one of the most ambitious comparative studies—in terms of the number of species that were used to test this way of understanding the evolution of cognition—showed that a single selective pressure (i.e., the social environment) does not correlate with behavioral performance on specific cognitive tests in birds and mammals, but does correlate with that of some groups [20]. This would indicate that the evolution of similar cognitive–behavioral phenotypes exhibited by different species would have occurred under dissimilar selective pressures.

These two views have very different consequences on interpreting cognitive evolution, and there has been no consensus regarding the discussion in question [17]. Neither of these two approaches seems to have transformed our understanding of cognition and its evolution [21] (Mercado III, 2024). As proposed by Gould [1] (1977), a possible third view appears from studies on the evolution of ontogeny.

#### 1.1.3. Slowed Development and Evolution of Cognitive Behavior

Despite their differing proposals on the explanation of the evolutionary process resulting in neoteny, both Lorenz [8] and Gould [1], along with many other authors such as Montagu [7], underscored the crucial role of slowed or retarded development in the evolution of human cognition. They also drew parallels between some notions of human adult cognition and the cognition of young animals, such as the presence of play behaviors in young animals and in young and adult humans. More recently, some other authors have also recognized the adaptive role of neoteny in the evolution of human cognition [22].

Evolutionary analysis implies recognizing the adaptive advantages of slowed development and its disadvantages or evolutionary costs. For instance, Bjorklund [22] pointed out that the physical dependence of human babies is much greater than that of other mammals because, at first, they must be carried and held by their mothers for an extended period. Even later, when they can move independently, they still do not possess the physical and mental skills necessary for independence. In other words, prolonging the infant and juvenile ontogenetic stages produced by the neotenic process in humans has implied the need for prolonged altricial care.

However, most authors agree that the selective costs involved in extending parental care are outweighed by the benefits or advantages offered by neoteny in terms of neural and behavioral plasticity, giving the possibility of increasing cognitive development, both by allowing a more extended safe period of non-social learning and by facilitating social learning [22,23].

The balance between the benefits of extended juvenile cognition and the costs of prolonged altricial care may have been an evolutionary pathway by which humans and other species of phylogenetically distant animal groups developed remarkable and similar cognitive abilities. Such is the case of the animal families Delphinidae [24], Psittacidae [25], Corvidae [26], Octopodidae [27], and Hominidae [28]. To support this claim, several comparative analyses in birds and mammals have shown that diverse developmental modes (on a continuum from precocial to altricial) are positively associated with both direct measures of cognition (e.g., performance on cognitive tests [29,30]) and indirect measures of cognition (e.g., brain size relative to body size [23], or size of the telencephalon relative to total brain mass [31]).

The comparative relationship between cognitive performance and the speed of development accompanied by the extended altricial condition favors the hypothesis advanced by Gould [1] that selective processes favor the extension of juvenile periods.

Cognitive processes, whether the so-called elementary—involving learning, memory, and emotions—or the so-called complex—involving thinking, reasoning, problem-solving, and decision-making [32]—are engaged in constructing behavioral phenotypes. The developmental processes that generate these cognitive–behavioral phenotypes must be carefully analyzed from the various aspects that we recognize (i.e., neural, neuronal, cognitive, and behavioral) [22].

### 1.2. Phenotypic Organization

#### 1.2.1. Phenotypic Plasticity

Phenotypic plasticity is a term used by scholars to refer to a phenomenon widespread in all living organisms [33]. Unlike the so-called rigid phenotypes that are thought to develop in the same way, independently of the environment, this phenomenon describes the ability of an organism to produce various traits depending on the environmental circumstances, some of which are thought to be persistent throughout life, while others are considered temporary adjustments [33]. The first type, known as the irreversible plastic phenotype (also known as the plastic phenotype and the developmental plastic phenotype [34]), involves the ability of developing organisms to produce different phenotypes in correspondence with the environment where development occurs [35]. The second type, known as the reversible plastic phenotype (also known as reversible plasticity, acclimation, activational plasticity, and contextual plasticity [34]), refers to situations in which an individual can, throughout their life, reversibly and repeatedly modify their phenotype in response to the conditions of their environment [34].

The notion of phenotypic plasticity is derived from the concept of phenotype, which, in turn, is related to the concept of genotype. How the relationship between the two concepts is conceived may vary between research programs and those who carry them out [15]. The distinction between these two entities is relevant because it defines the evolutionary thinking with which a problem is approached [15]. In the thinking that adheres to the evolutionary theory described by Huxley as the modern synthesis, evolutionary change occurs due to interactions at the organismal level between two types of processes. On the one hand, internal processes generate variability in an autonomous and alienated way from the living being [36,37]. On the other hand, external processes either facilitate or impede the reproduction of organisms according to the adaptability of the phenotypic traits they display in their development. In this view, genetic material is the primary source of interindividual variation, and it is the material that will be inherited [36]. Thus, for several decades, the relationship between genotype and phenotype was considered a linear process following a pre-established developmental program [36]. Since, in this view, alterations inherent to any developmental process do not affect evolutionary change, the study of the aforementioned relationship seemed irrelevant [36].

Here, a notion relating the two concepts in a processual sense is used [15] so that the genetic aspect of an organism is understood as an organizing principle of functional differentiation that operates in the phenotypic constitution of an organism, through the constant interaction with its environment from the beginning to the end of its development. This developmental process can be seen as a series of phenotypic ramifications that occur due to functional differentiations that take place at different times during the life of the organism and at different contextual points both inside the organism (for example, in the chromosomal or brain topographies) and outside it (for example, in various temperature and humidity conditions). This view follows the tradition of Lewontin’s processual thinking [36,38,39,40], named after the scientist considered to be one of the founders of what is known as the evo-devo approach [41,42], where genes cease to be the bearers of instructions for assembling a preconceived organism and become one more element among many others involved in a complex process of reciprocal definitions, ranging from the arrangement of genes in chromosomal spaces as random events that occur within the cellular space to the moment of development in which interactive contact occurs with some environmental facet, as well as the history, even intergenerational, of these occurrences [38,39]. Accordingly, the phenotypes of individuals are due to the biochemical activity of their genes in a unique sequence of environments and the development that can occur subsequently due to the initial action of those genes [39]. In more recent terms, phenotypic processes are built from reciprocal causalities in a tripartite manner between the genes, environment, and organism [40].

The phenomenon called phenotypic plasticity is relevant in the evo-devo evolutionary approach precisely because it is a developmental process by which organisms generate complementarity with their environment [43,44,45].

Phenotypic plasticity is often defined as a phenotypic change in individual organisms associated with different environments [42,46]. The term change is relevant here, as it implies a departure from an original plan (i.e., something that, due to its evolutionary history, must have had another form, but changed).

Based on the etymology of the term, plasticity has two meanings in biology. It can mean shaping or molding, or it can refer to the existence of an alternative [47]. This distinction has consequences for understanding phenotypic processes, which are multiplied if we add the temporal factor.

The second meaning of the term plastic refers to the notion of switching between alternative results depending on the environment, which, in turn, is consistent with the idea of genetically pre-established phenotypic possibilities that are each associated with the environmental conditions encountered in the past by previous generations [48]. Contemporary studies on phenotypic plasticity acknowledge that phenotypes result from the interaction between genes and the environment that occurs during the life of an organism [44,48]. However, even in this perspective, where gene expression is inherently context-dependent, the concept of a plastic phenotype as an expected expression of the genotype sometimes persists [48].

Similarly, the notion of plasticity as the ability to shape can be understood as the ability of the environment to mold the phenotype through generations, or as the ability of the body to mold itself over its life in correspondence to the environment. Considering the advances in evolutionary theory and the consequent necessity to focus on development [41,42], the notion of a developing organism able to mold itself is in line with the fact that several phenotypic expressions are influenced by inherited environmental information, immediate epigenetic modifications of the genome, and conditions encountered previously within the lifecycle [46]. Therefore, phenotypic variations can be conceived as “unscripted” developmental responses to environments [48], and the phenomenon called phenotypic plasticity can be defined as a developmental process that generates complementarity between the organism and its environment [43].

These considerations indicate that all types of phenotypes (i.e., fixed, plastic, and reversible) are the result of contextualized developmental processes; so the term plasticity, in the sense that it is the ability of the developmental organism to mold itself, can be applied to all of them. Perhaps it would be advisable to review its use. Either way, understanding phenotypes as developmental processes provides a consistent approach with a constructive vision of the role of behavioral phenotypes in establishing a correspondence between the organism and its environment.

In several animal species, the phenomenon known as phenotypic plasticity is more frequent in the initial stages than in the later stages of the lifecycle [49,50].

#### 1.2.2. Behavioral Phenotypic Plasticity

Behavioral phenotypic plasticity is defined as the ability of a genotype to produce different behaviors across environments [51]. Behavioral phenotypic plasticity allows organisms to adjust their behavior to variable situations, helping them to dovetail better into various environments and circumstances. This phenomenon includes at least five aspects of phenotypical plasticity (i.e., morphological plasticity, physiological plasticity, neural plasticity, cognitive plasticity, and adaptive cultural plasticity) [52]. Two types of behavioral phenotypic plasticity have been distinguished: developmental and activational [51]. Developmental–behavioral phenotypic plasticity can be tied to different developmental trajectories deployed in various environments. In this definition, developmental–behavioral phenotypic plasticity encompasses the morphological, physiological, and neural aspects relevant to a particular behavior. Instead, activational–behavioral phenotypic plasticity is an immediate response to the environment, referring to the differential activation of an underlying neuronal and muscle network [51]. The distinction between the two types of behavioral phenotypic plasticity rests in the process associated with the neural network. The developmental type refers to the conformation of the network, while the activational refers to the initiation of those networks. There is a difference in the time scale.

As in the notion of plasticity described above [47], behavioral phenotypic plasticity also has two meanings, affecting our understanding of this phenomenon. When the notion of plasticity is conceived as an alternative, it implies a connection between the purpose of the behavior and the genetic makeup of the organism. This means understanding the evolution of behavior as the process by which several adaptive solutions to specific environmental problems are fixed in the genetic baggage of organisms. In this view, the organism must be prepared to receive and decode the environmental cues to adjust to its environment.

However, the production of behavioral phenomena seems more complicated than what was thought decades ago, particularly regarding the role of genetic material in this process [53,54].

The generation of behavior implies not only the formation of a movement but also the elaboration of a specific objective. Behavior is an integral part of cognition. It serves the functions of expressing, carrying out, acquiring, and modulating the flow of information between the organism and its environment and adjusting internal states [55]. In terms of behavior, intentions and plans are carried out rather than muscular movements, and consequently, strictly speaking, what is observed is not movements but actions. The same action can be carried out with different movements, and the same movement can be used to execute other actions. Actions are structured by perception: they are cognitive [55]. Accordingly, developing a behavioral phenotype involves coordinating internal sensations, body awareness, and perception of the external environment to establish a goal and carry out a movement. Forming a behavioral phenotype involves incorporating several cognitive processes, such as interoception, proprioception, and exteroception, to calibrate a movement with the goal in real-time. Behavioral phenotypes depend on other phenotypes, such as neural and cognitive [51]. The goal of a particular behavior must be considered a property of the interaction between the organism and its environment; therefore, it cannot be a property of the genotype, but rather, it is a property of the phenotype. Importantly, this link is distinct from genetic assimilation or epigenetics [56] because each new interaction between the genotype and environment leads to a unique phenotypic outcome, creating a “genotype–environment entanglement” [48].

To define the behavioral phenotype, it is necessary to separate the purpose of the behavior from its genetic aspect. The developing organism is not searching for environmental cues but is shaping itself by interacting with the environment. In other words, environmental cues are not there, but the organism senses the environment to integrate the phenotype accordingly. Therefore, the processual definition of phenotypic plasticity mentioned earlier [43] can be expanded to include any other developmental process that creates complementarity between the organism and its environment through behavior. Following the embryological tradition, the first meaning of plasticity already mentioned [47,57] suggests it is a system that shapes itself by interacting with the environment. Thus, any behavior can be seen as the result of a developmental phenotype.

#### 1.2.3. Neural Phenotypic Plasticity

The term neural plasticity refers to the ability of organisms to create and adjust the structure and function of their neural networks in response to their environment or to a specific neural lesion or accident [58]. Plasticity occurs at different organizational levels of the nervous system. Thus, we can speak of nervous tissue plasticity, neuronal or glial plasticity, synaptic plasticity, etc. [58]. Neural plasticity can be structural and functional [52]. In humans, these processes are crucial for learning from experience, forming memories, and other cognitive and behavioral functions. While particularly important in early life, the possibility of neural plasticity continues into old age. The brain consistently reorganizes and reshapes itself through interactions with the environment and peripheral organs [57].

Like any other organ, the brain’s formation begins as a process of cellular differentiation. In a multicellular organism, cell differentiation depends on its context. Once this differentiation has taken place, the activity of the cell is determined. In the nervous system, neurons use these external regulatory systems to create synapses [59]. Creating synapses is how organisms use experience to act [59]. At early stages, brain development follows a sequence of events remarkably conserved across species [60]. However, the duration and speed of the unfolding processes vary across and within species. They also vary among brain regions and cell identities [60,61]. As maturation progresses, neurons grow, increasing their size, morphological complexity, excitability, and connectivity [60]. It is important to note that epigenetic pathways regulate neural events in the brain and other parts of the central nervous system throughout the lifecycle [60,62]. Human brain development, particularly its neuronal maturation, is prolonged compared to other species [60]. Neural phenotypic plasticity is fundamental to brain development [63].

The brain exhibits various types of phenotypic plasticity based on its durability. Some electrophysiological states are fleeting, while others are a permanent part of their anatomical structure. Neural plasticity can lead to anatomical changes at different levels. For instance, generating new connections between neurons (synaptic plasticity) enables the early development of neuronal networks, which can later be refined through experience or altered due to injury [64]. Two additional forms of plasticity occur in the adult brains of various species, including humans. These are the functional differentiation of neurons from stem cells (adult neurogenesis) and the functional differentiation of neurons from immature cells (cortical immature neurons) [64]. These two cellular differentiation processes vary widely across vertebrate species regarding their developmental trajectories, but both are believed to play a dominant role in cognitive processes [64].

These neural aspects are related to human neoteny, and human synaptic neoteny has been linked to the evolution of human cognition [65].

#### 1.2.4. Cognitive Phenotypic Plasticity

Neural and behavioral phenotypic processes have cognitive aspects. Cognition is a generic term referring to various phenomena that, in turn, consist of multiple processes [11]. Accordingly, there are two ways of understanding cognitive processes. One view suggests that cognitive processes involve reasoning, that they operate with propositions (i.e., statement-like representations), and that they involve desires, beliefs, and other intentional mental states; furthermore, these processes could be available to consciousness. The other view suggests that a process is cognitive when it adaptively uses information and can be modeled as an algorithm [11]. The latter definition aligns better with the content of this manuscript.

A cognitive phenotype is an informative solution to a problem devised by appropriately deploying relevant knowledge made available through interoception, proprioception, and exteroception. Cognitive phenotypic plasticity refers to information processes (i.e., cognitive phenotypes) produced according to different percepts [66].

Animal cognition is shaped through individual experience. A striking case of this phenomenon is observed when comparing individuals raised in impoverished environments with individuals raised in enriched environments. Enriched environments imply changes in the brain and improve cognitive abilities in various species. For example, guppy fish that were raised in a fish tank with other individuals, gravel on the bottom, natural and artificial plants, and live prey showed better performance in tests of learning (i.e., where the individual must learn to discriminate a color to obtain a reward), reversal learning (i.e., where the individual has to choose the color option that was not rewarded in the learning test), and self-control (i.e., where the individual must refrain from attacking prey that is behind a transparent barrier) compared to individuals that were raised in impoverished environments [67].

There is evidence that cognitive performance is positively associated with levels of neurogenesis in the hippocampus [68]. In mice, cognitive decline associated with aging involves a reduced ability to adjust previous experiences to solve tasks involving new combinations of familiar contexts and cues, and it is correlated with decreased numbers of mature neurons or newly generated immature precursors in the hippocampus [68].

## 2. From Types of Phenotypic Plasticity to Aspects of Phenotypic Processes of Differentiation

The activational phenotype mentioned earlier [51] can be seen as a step in differentiating a specific function in phenotypic terms (i.e., neural, cognitive, or behavioral). At the beginning of the differentiation process, the phenotypes still correspond to multiple components of the environment and not just one. As the differentiation process progresses, the phenotypes become specified.

Think, for instance, of the moment in which two people meet each other for the first time and their subsequent encounters. At the first interaction with someone new, a neural network is activated in the brain. This activation can happen when the face of this person is seen, her name is heard, or when engaging with her by any other sensory modality or by two or more modalities combined in any way. At this moment in the interaction, the precision of the network concerning the identity of that person depends on the categories already formed in the brain and the possibility of including the person in those categories from what is being perceived, such as whether the person is a colleague or a teacher or if they are older or younger. During this initial encounter, the ongoing cognitive process of fencing, or reducing the possibilities of action toward the new acquaintance, depends on the cognitive phenotypes already formed in the brain. The various neural networks that can be activated are not specific to that person, so the resulting actions, such as a polite greeting, are impersonal. However, in later encounters, the person will be recognized individually, guiding the behavior toward her in a more specific and familiar manner. Processes of this kind involve many phenotypic terms, such as neural or cognitive, which are dependent on sleep [69]. Similar processes can occur with other components of the environment, like physical objects, chemical properties, prey, predators, and many more. The cellular systems that correspond to these processes are very sophisticated and contain diverse elements (e.g., associative, inhibitory, and excitatory) organized hierarchically [70].

Individual recognition is a crucial cognitive ability for social interactions in various animal groups, including vertebrates like fish, birds, and mammals and invertebrates like arthropods. This ability is essential for proper behavior in diverse social contexts, like cooperation, competition, and parenting. Regardless of the sensory modality each group uses (e.g., chemical, visual, or auditory), their neural systems must be able to distinguish other individuals [71]. Facial recognition has been widely studied in humans, with impressive results. For instance, Quiroga and colleagues found that a specific region of the brain contains neurons selectively activated by different instances of a well-known actress, whether in the form of her picture, a drawing, or her written name [72]. The authors were cautious in interpreting the data but suggested that future research might reveal an explicit and invariant encoding of visual percepts [72]. Moreover, some studies indicate that new neurons added to an adult brain could replace older neurons or provide additional cells to expand an existing network [73]. With this evidence in mind, immature neurons can likely be utilized to develop the finest phenotypes needed to specify the interaction with the environment, either neuronal, cognitive, behavioral, or other.

Recently, Gordon and colleagues [74] generated a series of high-fidelity, highly individual-specific functional connectomes. This is a much less invasive neuroimaging technique than the one used by Quiroga and colleagues and also provides evidence that neural phenotypes are generated and specified through interaction with the environment. Their approach revealed interindividual variability in the spatial and organizational distribution of brain networks associated with the cognitive tasks performed by the individuals studied. The authors reflected that these interpersonal variations in connectomes may be related to demographic, cognitive, or personal differences [74].

There is also evidence that facial recognition in other animals occurs through environmental interaction. For example, in the case of two species of the same genus of wasps, geographic variation in individual face learning is not based on genes but on social experience: the individual learns to distinguish and remember the unique phenotypic features of another individual’s face and to associate them with aggressive and/or affiliative social behaviors [72].

## 3. Discussion

This essay presents an explanatory hypothesis about the relationship between neural differentiation and the generation of a novel behavioral phenotype, a topic that is inherently complex and multifaceted. This hypothesis, which includes neuronal, neural, cognitive, and muscular aspects, is a testament to the intricate nature of phenotypic development. It is important to note that, in this view, elaborating the specific goal of this behavioral phenotype (i.e., its purpose) requires interaction between the organism and its environment. Therefore, sensorial, perceptual, cognitive, or behavioral constructions are considered phenotypic properties. The starting point is that every phenotype is plastic since its generation is environmentally contextualized. Furthermore, the interactions between the organism and its environment (whether material or mental) can occur from already-established behavioral phenotypes (i.e., previously generated in the life of the organism). These interactions will reach a certain degree of generic complementarity but not all, as some interactions require specificities (for example, social interactions). The generation of a more specialized behavioral phenotype is slower than the generation of a more generalized behavioral phenotype. Generating a new behavioral phenotype that increases complementarity with the environment requires the activation of previously established neural networks (ranging, for example, from sensorimotor structuring to the formation of a percept and an intention). Still, new neural units are required to form a more specialized network. In this sense, novel behavioral phenotypes imply the functional differentiation of new cells. Greater behavioral specificity (for example, behaving with a person uniquely depending on their recognition and the circumstance in which the interaction is occurring) is acquired by forming a neural network with a distinctive element of the percept.

The prolongation of the infantile and juvenile stages produced by the neotenic process in humans has implied the need for prolonged altricial care. Consequently, it has allowed the prolongation of the creation of highly differentiated behavioral phenotypes into adulthood. This process has not been exclusive to the human species. It may have been the evolutionary solution to the need to develop remarkable cognitive capacities in various phylogenetically distant animal species, since these modes of development show positive correlative evolution with direct and indirect measures of cognition. It seems feasible to speculate that in species with slowed or retarded development compared to the speed of development in ancestor species (i.e., neotenous), new, highly specialized behavioral phenotypes can be created through neural differentiation during most of the lifecycle. This hypothesis focuses particularly on behavioral phenotypes (and their corresponding neural and cognitive phenotypes) but can be generalized to any phenotype (i.e., physiological, hormonal, morphological, etc.).

## 4. Conclusions

Human development has slowed compared to that of the great apes, allowing the retention of juvenile traits into adulthood. These neotenic characteristics are reflected in several aspects. It has been hypothesized that the cognitive–behavioral aspects of neoteny were crucial in the evolution of human development. Comparative studies that show a positive correlation between cognitive–behavioral performance and rearing time support this hypothesis.

The study of the organization of cognitive–behavioral phenotypes is important to broaden our understanding of the evolution of human behavior. The analysis of some phenotypic concepts supports the conclusion that all phenotypes—rigid, plastic, and reversible—are plastic; therefore, the author suggests reconsidering the use of this adjective to distinguish phenotypic processes. The genetic aspect of an organism is understood as an organizational principle of functional differentiation that operates in the phenotypic constitution of an organism through constant interaction with its environment, from the beginning to the end of its development, supporting the suggestion that the processes of phenotypic construction and differentiation are linked.

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
