# Peer review of "Reflections on the Role of Differentiation Processes in Forming Behavioral Phenotypes: Can These Processes Replace the Concepts of Plastic Phenotype and Reversible Plastic Phenotype?"

_biology, 2025, doi:10.3390/biology14020187_

Round 1

Reviewer 1 Report

Comments and Suggestions for Authors

In this essay, the author discussed the notions of plastic phenotype and reversible plastic phenotype, and the processes involved in developing phenotypes based on comparative analyses of the correlative evolution of developmental and cognitive processes. Based on the positive correlative evolution between cognitive behavior and modes of development, the author argued in favor of replacing the concept of behavioral plasticity with cognitive differentiation. However, there are certain parts that are unclear and should be rephrased or reorganized.

1.      The author mentioned the classical concept of behavioral plasticity and the conceptual contrast; however, the author did not come to a clear conclusion from his/her analysis and comparison studies about which concept is more reasonable.

2.      In the essay, the author proposes a few arguments, but the support and evidence behind the arguments are not clearly elucidated. For example, in lines 303-326, the author proposed one argument, but the reasoning in this paragraph is a bit confusing. One issue is that certain concepts were brought up repeatedly instead of using one clear and concise description. The author should improve the writing and organization throughout the paper.  

3.      The language is not professional and too extreme in certain parts. For example, in the conclusion, the author mentioned that “Describing a phenotype with these adjectives 425 is useless” without any further elaboration.

Comments on the Quality of English Language

The scientific writing and arguments of this paper should be improved. 

Author Response

Response to Reviewer 1 Comments

1. Summary

Thank you very much for taking the time to review this manuscript. Please find the detailed responses below and the corresponding corrections in the re-submitted files.

2. Questions for General Evaluation

Reviewer’s Evaluation

Response and Revisions

Does the introduction provide sufficient background and include all relevant references?

Can be improved

I changed the introduction by including the references another reviewer advised me to include. I hope these changes have improved this section.

Is the research design appropriate?

Can be improved

Indeed, the essay had serious problems with its argumentative structure. I hope that these problems have been corrected in this new version.

Are the methods adequately described?

Can be improved

I assumed this comment was about the logical structure of the essay. I changed the order of the arguments presented, trying to achieve a more logical format. However, I am not sure that I achieved that goal.

Are the results clearly presented?

Must be improved

I improved my argument and presented it more formally. I hope succeeded.

Are the conclusions supported by the results?

Must be improved

I modified the final part of the essay, expanding the discussion and removing the conclusions.

3. Point-by-point response to Comments and Suggestions for Authors

Comments 1: The author mentioned the classical concept of behavioral plasticity and the conceptual contrast; however, the author did not come to a clear conclusion from his/her analysis and comparison studies about which concept is more reasonable.

Response 1: Thank you for pointing this out. I agree with this comment. Therefore, I integrated other phenotype concepts and added a conclusion to the review of these concepts.

Comments 2: In the essay, the author proposes a few arguments, but the support and evidence behind the arguments are not clearly elucidated. For example, in lines 303-326, the author proposed one argument, but the reasoning in this paragraph is a bit confusing. One issue is that certain concepts were brought up repeatedly instead of using one clear and concise description. The author should improve the writing and organization throughout the paper.

Response 2: I completely agree. I used a definition for each concept and improved the support for the proposed argument. I changed the wording and structure a lot, so the lines no longer correspond and I cannot refer to them explicitly here. However, the changes that best correspond to this commentary can be found in lines 399-452.

Comments 3: The language is not professional and too extreme in certain parts. For example, in the conclusion, the author mentioned that “Describing a phenotype with these adjectives 425 is useless” without any further elaboration.

Response 2: I apologize for the arrogance I displayed in the language used in the last version. The language has been changed, using a formal style.

4. Response to Comments on the Quality of English Language

Point 1: The scientific writing and arguments of this paper should be improved.

Response 1: The scientific writing and arguments were improved.

5. Additional clarifications

I found the comments to be very accurate. I had a hard time meeting the deadlines assigned to me, mainly due to the depth of the corrections.

Reviewer 2 Report

Comments and Suggestions for Authors

GENERAL COMMENT:

This contribution triggers a double, somehow discordant, judgment. i) It definitely regards an important, often forgotten issue, i.e., the neotenic processes very likely involved in the evolutionary appearance of Homo on our planet, and does it in a (partially) appropriate framework. Yet (ii) its expositive style is strictly remembering of an oral narrative, a kind a lecture-type speech that can result rather bizarre to the mainstream reader of a typical scientific journal. Finally, the cultural role of the late geneticist and militant anti-reductionistic scientist Richard L. Lewontin (Harvard Un.) should be introduced and discussed in this paper. 

SPECIFIC POINTS:

Introduction 

Pag 2 Lines 59-61 

In this sense, the earliest adaptive 59 function of human neoteny is that of Konrad Lorenz, presented in 1950 in his essay “Part 60 and Parcel in Animal and Human Societies. A methodological discussion.” [7] 

and

Lines 76-78 

Decades later, Stephen Jay Gould also attributed an essential role to heterochrony in human evolution. However, he shifted the focus of selective forces from morphological to behavioral: “Neoteny has been a (probably the) major determinant of human evolution. 

I do appreciate the primigenious, often forgotten, input released by Lorenz. However, It seems to me that Gould’s role (he was a Pulitzer Prize recipient, after all) consisted more in diffusing and broadcasting point of views of other authors having postulated those neotenic processes leading to Homo that in his own research lines.

Lines 125-127 

Phenotypic plasticity is often defined as a phenotypic “change” in individual organisms associated with different environments [20,21]. The term “change” is significant here, as it implies a departure from an original plan (i.e., something that supposedly had to 

R.L. Lewontin’s book “The genetic basis of the evolutionary change” Columbia UP, 1974 had a pivoltal role in “modernizing” the evolutionary CHANGE as postulated by Darwin himself. It should be attentively discussed here.

Editore, Columbia Univ Pr (1 settembre 1974).

Lines 277-279 

Emilie Snell-Rood [33] distinguished two types of behavioral phenotypic plasticity: 277 developmental and activational. As I understand her approach, developmental-behav- 278 ioral phenotypic plasticity can be tied to different developmental trajectories deployed in 

Is a typical scientific paper published in a periodical, the expression “As I understand” sounds definitely bizarre and misleading.  It also reveals the “oral speech” style permeating this present contribution. It may trigger some doubts about its  scientific weight in a reader prepared for years to a more conventional language.

4. Discussion 

Lines 376-380 

In this essay, I argue that the notions of irreversible plastic phenotype and reversible 377 plastic phenotype are incompatible with the current knowledge about the processes in- 378 volved in developing phenotypes. I also say that all phenotypes are plastic and that re- 379 versibility has not been demonstrated; therefore, I maintain that describing a phenotype 380 

In some measure the “current knowledge” the Author refers to is not sufficiently delineated in the paper. An effort to do so will be welcomed.

5. Conclusions 

Lines 422 -425 

The notions of irreversible and reversible plastic phenotypes are incompatible with 423 current knowledge about the processes involved in developing phenotypes. All pheno- 424 types are plastic, and they are not reversible. Describing a phenotype with these adjectives 

The present Conclusion enlists a series of statements already reported in the Discussion, with about the very same wording. It results more a kind of Abstract, therefore. An effort to make it “a step further” for the reader may be attempted.

Author Response

Response to Reviewer 2 Comments

1. Summary

Thank you very much for taking the time to review this manuscript. Please find the detailed responses below and the corresponding corrections in the re-submitted files.

2. Questions for General Evaluation

Reviewer’s Evaluation

Response and Revisions

Does the introduction provide sufficient background and include all relevant references?

Can be improved

I changed the introduction by including the references that you advice me. I hope these changes have improved this section.

Is the research design appropriate?

Not applicable

Are the methods adequately described?

Not applicable

Are the results clearly presented?

Not applicable

Are the conclusions supported by the results?

Must be improved

I modified the essay's final part, expanding the discussion and removing the conclusions.

3. Point-by-point response to Comments and Suggestions for Authors

This contribution triggers a double, somehow discordant, judgment. i) It definitely regards an important, often forgotten issue, i.e., the neotenic processes very likely involved in the evolutionary appearance of Homo on our planet, and does it in a (partially) appropriate framework. Yet (ii) its expositive style is strictly remembering of an oral narrative, a kind a lecture-type speech that can result rather bizarre to the mainstream reader of a typical scientific journal. Finally, the cultural role of the late geneticist and militant anti-reductionistic scientist Richard L. Lewontin (Harvard Un.) should be introduced and discussed in this paper.

Comments 1:

Pag 2 Lines 59-61

In this sense, the earliest adaptive 59 function of human neoteny is that of Konrad Lorenz, presented in 1950 in his essay “Part 60 and Parcel in Animal and Human Societies. A methodological discussion.” [7]

And

Lines 76-78

Decades later, Stephen Jay Gould also attributed an essential role to heterochrony in human evolution. However, he shifted the focus of selective forces from morphological to behavioral: “Neoteny has been a (probably the) major determinant of human evolution.

I do appreciate the primigenious, often forgotten, input released by Lorenz. However, It seems to me that Gould’s role (he was a Pulitzer Prize recipient, after all) consisted more in diffusing and broadcasting point of views of other authors having postulated those neotenic processes leading to Homo that in his own research lines.

Response 1: I agree partially with this comment. I included Montagu as another of the main authors who recognized the fundamental role of neoteny in human evolution. However, I must reaffirm that I am not aware of any reference prior to Gould's 1977 Ontogeny and Phylogeny where the hypothesis is made that it was the cognitive-behavioral characteristics of human neoteny that were selected for. If you could suggest one, I would be happy to add a new author in this regard.

Comments 2:

Lines 125-127

Phenotypic plasticity is often defined as a phenotypic “change” in individual organisms associated with different environments [20,21]. The term “change” is significant here, as it implies a departure from an original plan (i.e., something that supposedly had to

R.L. Lewontin’s book “The genetic basis of the evolutionary change” Columbia UP, 1974 had a pivoltal role in “modernizing” the evolutionary CHANGE as postulated by Darwin himself. It should be attentively discussed here.

Editore, Columbia Univ Pr (1 settembre 1974).

Response 2: I completely agree. I have reread several texts by R. Lewontin, including the one you suggested. I have incorporated several quotes from this author. I hope I have met your expectations.

Comments 3:

Lines 277-279

Emilie Snell-Rood [33] distinguished two types of behavioral phenotypic plasticity: 277 developmental and activational. As I understand her approach, developmental-behav- 278 ioral phenotypic plasticity can be tied to different developmental trajectories deployed in

Is a typical scientific paper published in a periodical, the expression “As I understand” sounds definitely bizarre and misleading.  It also reveals the “oral speech” style permeating this present contribution. It may trigger some doubts about its  scientific weight in a reader prepared for years to a more conventional language..

Response 3:

I apologize for the arrogance I displayed in the language used in the last version. The language has been changed, using a formal style.

Comments 4:

Discussion

Lines 376-380

In this essay, I argue that the notions of irreversible plastic phenotype and reversible plastic phenotype are incompatible with the current knowledge about the processes in-volved in developing phenotypes. I also say that all phenotypes are plastic and that reversibility has not been demonstrated; therefore, I maintain that describing a phenotype

In some measure the “current knowledge” the Author refers to is not sufficiently delineated in the paper. An effort to do so will be welcomed.

Response 4: I agree. I have substantially modified the essay in order to improve it, both in the clarification of concepts and in the argumentative structure. I hope I have achieved this reasonably well.

Comments 5:

Conclusions

Lines 422 -425

The notions of irreversible and reversible plastic phenotypes are incompatible with 423 current knowledge about the processes involved in developing phenotypes. All pheno- 424 types are plastic, and they are not reversible. Describing a phenotype with these adjectives

The present Conclusion enlists a series of statements already reported in the Discussion, with about the very same wording. It results more a kind of Abstract, therefore. An effort to make it “a step further” for the reader may be attempted.

Response 5: I agree. I'm considering removing the conclusions to make the essay more coherent. I hope this action wasn't too radical.

4. Response to Comments on the Quality of English Language

The quality of English does not limit my understanding of the research.

Response 1: The scientific writing and arguments were improved.

5. Additional clarifications

I found the comments to be very accurate. I had a hard time meeting the deadlines assigned to me, mainly due to the depth of the corrections.

Round 2

Reviewer 1 Report

Comments and Suggestions for Authors

The previous comments were addressed and no more other comments.